# Neobavaisoflavone Inhibits Melanogenesis through the Regulation of Akt/GSK-3β and MEK/ERK Pathways in B16F10 Cells and a Reconstructed Human 3D Skin Model

**DOI:** 10.3390/molecules25112683

**Published:** 2020-06-09

**Authors:** Da Eun Kim, Bo Yoon Chang, Sang Ok Ham, Youn Chul Kim, Sung Yeon Kim

**Affiliations:** 1Institute of Pharmaceutical Research and Development, College of Pharmacy, Wonkwang University, Iksan 54538, Korea; dhrtls1234@naver.com (D.E.K.); oama611@nate.com (B.Y.C.); 2Yougwon Makers CO., LTD., 312, 1646, Yuseong-daero, Yuseong-gu, Daejeon 34054, Korea; sohalm94@gmail.com; 3College of Pharmacy, Wonkwang University, Iksan 54538, Korea; yckim@wku.ac.kr

**Keywords:** *Pueraria lobata*, neobavaisoflavone, anti-melanogenesis, 3D human skin model, tyrosinase, Akt/GSK-3β, MEK/ERK

## Abstract

Previous studies have confirmed the anti-melanogenic effect of the aerial part of *Pueraria lobata*, however, due to its inherent color, *P. lobata* has limited commercial use. In this study, an extract (GALM-DC) of the aerial part of *P. lobata* having improved color by the use of activated carbon was obtained. Furthermore, the active compound neobavaisoflavone (NBI) was identified from GALM-DC. The effect of NBI on melanogenesis, tyrosinase activity, α-glucosidase activity, and mechanism of action in melanocytes was investigated. Tyrosinase activity, melanin contents and the expression of melanin-related genes and proteins were determined in B16F10 cells. NBI reduced melanin synthesis and tyrosinase activity. Furthermore, NBI treatment reduced the mRNA and protein expression levels of MITF, TRP-1, and tyrosinase. NBI also works by phosphorylating and activating proteins that inhibit melanogenesis, such as GSK3β and ERK. Specific inhibitors of Akt/GSK-3β (LY294002) and MEK/ERK (PD98059) signaling prevented the inhibition of melanogenesis by NBI. NBI inhibited melanin production through the regulation of MEK/ERK and Akt/GSK-3β signaling pathways in α-MSH-stimulated B16F10 cells. NBI suppresses tyrosinase activity and melanogenesis through inhibition of α-glucosidase activity. Besides, NBI significantly reduced melanogenesis in a reconstructed human 3D skin model. In conclusion, these results suggest that NBI has potential as a skin-whitening agent for hyperpigmentation.

## 1. Introduction

Melanin is produced from melanocytes secreted between the epidermis and the dermis. Melanin is a phenolic polymer widely distributed in plants and animals, and a major factor determining the color of the skin and hair functions to inhibit damage to skin cells by external stimuli, such as ultraviolet (UV) light and free radicals [1,2]. When melanin is over-synthesized, it is deposited on the surface of the skin, causing various pigmentations, such as spots and freckles, and promoting skin aging [3].

Melanin synthesis is stimulated by several agents and conditions, including α-melanocyte stimulating hormone (α-MSH), isobutyl methylxanthine (IBMX), and UV radiation. Dermal hyperpigmentation may depend on increased numbers of melanocytes or increased melanogenic enzyme activities [4,5]. The first two steps in melanin synthesis, from L-tyrosine to 3,4-dihydroxy-L-phenylalanine (L-DOPA to L-DOPA quinone), are catalyzed by tyrosinase, and the remaining steps are catalyzed by several different enzymes, including tyrosinase-related proteins (TRP-1 and TRP-2) and tyrosinase itself. Therefore, inhibition of tyrosinase, TRP-1, and TRP-2 activity is important for skin whitening [6,7]. Microphthalmia-associated transcription factor (MITF) is an important transcription control factor for genes responsible for melanin biosynthesis [8]. MITF is involved in the survival, proliferation, and differentiation of melanocytes. MITF regulates the expression of genes encoding melanin-producing enzymes, including TRP-1 and TRP-2 [9].

Researchers have developed melanin synthesis inhibitors, such as hydroquinone, arbutin, kojic acid, sulfite and azelaic acid. However, whitening agents based on tyrosinase inhibitors have serious side effects, such as cytotoxicity, an unfavorable odor, and instability in the presence of water and oxygen [10,11,12,13]. For this reason, studies on the development of whitening materials using medicinal plants and natural substances, which have little side effects on the human body, have been actively conducted.

*Pueraria lobata* is a perennial plant of the Fabaceae family. It is widely distributed in temperate East Asia [14]. *P. lobata* has been implicated as an important medicinal plant and is known for its anti-pyretic and hypotensive actions, its effect in menopausal and cardiovascular diseases and its protective effect against lead toxicity [15,16,17]. In recent studies, the aerial part of *P. lobata* has been reported to exert protective effects on the liver and to prevent bone disease and is therefore regarded as a highly useful material [18,19,20,21].

Previously, we showed that extracts from the aerial part of *P. lobata* contain isoflavonoids—including daidzin, daidzein, genistin, genistein, and puerarin-and possess anti-melanogenesis efficacy [22]. However, due to the inherent color of the aerial parts of *P. lobata*, there is a limit to its potential utilization in the cosmetic industry. Therefore, in the present study, we decolored the aerial parts of *P. lobata* using activated carbon.

Extracts (GALM-DC) from the decolorized aerial part of *P. lobata* were found to contain neobavaisoflavone (NBI, Figure 1) and seven other compounds, including betulinic acid, corylin, diadzein, puerarone, and 8-prenyldaidzeine. Among them, betulinic acid is known to be effective in inhibiting melanogenesis [23]. Antioxidants are well known to play an important role in the inhibition of melanogenesis [24,25]. NBI has been reported to have antioxidant, anti-tumor, hepatoprotective effect, platelet aggregation inhibition, and stimulates osteogenesis properties [26,27,28,29,30,31]. Based on NBI’s antioxidant properties, it was hypothesized that it could effect the inhibition of melanogenesis. Also, the effect of NBI on melanin production has not been studied.

This study aims to investigate the anti-melanin activity and mechanism of GALM-DC and GALM-DC-derived NBI using a reconstructed human 3D skin model with B16F10 cells.

## 2. Results

### 2.1. The Aerial Part of P. lobata Inhibits Melanogenesis in B16F10 Cells

After treatment with GALM-DC at 10, 25, 50, 100, and 200 µg/mL for 48 h, cell viability was measured by an MTT assay. As shown in Figure 2A, GALM-DC was not cytotoxic at concentrations ranging from 10 to 50 µg/mL. B16F10 cells were then treated with a non-toxic concentration of GALM-DC (10–50 µg/mL). As shown in Figure 2B,C, melanin contents and cellular tyrosinase activity were dose-dependently decreased following exposure to GALM-DC.

### 2.2. Antioxidant Activities of GALM-DC

The anti-oxidant effect of GALM-DC was determined by measuring its DPPH and SOD radical scavenging activities. DPPH radical scavenging activity at 100 and 500 µg/mL was 21.7 and 63%, respectively. The DPPH scavenging activity of 50 µM vitamin C (positive control) was 80.1% (Figure 3A). The SOD activities of GALM-DC at 100 and 500 µg/mL were 13.1 and 36%, respectively, while that of 500 µg/mL Trolox (positive control) was 50.4% (Figure 3B). All markers showed greater anti-oxidant activity at higher dosages, suggesting that GALM-DC possesses dose-dependent anti-oxidant activity.

### 2.3. Effects of NBI on Anti-Melanogenesis in B16F10 Cells

As NBI showed no cytotoxicity at 50 µM in B16F10 cells (Figure 4A), NBI was thus treated with 2–50 µM.

As shown in Figure 4B, the α-MSH treated group displayed increased melanin production compared to control. Arbutin (100 µM) was used as a positive control and 25.8% of melanin production was suppressed. NBI decreased the melanin production increased by α-MSH in a concentration-dependent manner. Ten and 50 µM NBI inhibited melanin production by 6.4 and 57.8%, respectively. The cellular tyrosinase inhibition rates of NBI were 10% and 45.5% at 10 and 50 µM, respectively, compared to the α-MSH treated group. The arbutin group also inhibited the cellular tyrosinase rate by 12% (Figure 4C). NBI therefore showed more potent anti-melanin activity than arbutin.

### 2.4. Inhibitory Effect of NBI on Glucosidase Activity

Tyrosinase is produced by a N-linked glycosylation process [32]. α-Glucosidase is one of many enzymes involved in the glycosylation process. When α-glucosidase is inhibited, the structure of tyrosinase is transformed and it migrates to melanosomes in an inactive form, resulting in inhibition of melanogenesis. Inhibition of glycosylation inhibits proper tyrosinase maturation and subsequent enzymatic activity [33,34]. In order to confirm the inhibition of the α-glucosidase activity of NBI, it was tested in a cell-free system. Acarbose, a potent α-glucosidase inhibitor, was used as a positive control [35]. Briefly, NBI was found to inhibit α-glucosidase more effectively than acarbose (Figure 5). Based on the calculated IC_50_ value, NBI reduced α-glucosidase activity with an IC_50_ of 554.3 µM (161.2 µg/mL) which is approximately 10-times lower than that of acarbose, at 1170.6 µg/mL.

### 2.5. Effects of NBI on Melanogenesis Signaling Pathways in B16F10 Cells

To determine the effect of NBI on the melanogenesis signaling pathway, cells were treated with NBI in the presence of α-MSH. The mRNA and protein expression levels of TYR, TRP-1 and MITF were then examined, respectively. Expression of MITF, TRP-1 and tyrosinase was increased by 100 nM α-MSH treatment. The inhibition rates of MITF expression were 19.7, 28.3, and 47% at 10, 25, and 50 µM, respectively, compared to the α-MSH treatment group. At 25 and 50 µM of NBI, TRP-1 expression was inhibited by 12 and 15.1%, and tyrosinase expression was inhibited by 20.8 and 23.9%, respectively. The transcription of MITF, TRP1, and tyrosinase was significantly attenuated in a dose-dependent manner (Figure 6A–C). As demonstrated in Figure 6D, the protein levels of MITF, TRP1, and tyrosinase were reduced by NBI compared to the α-MSH group. These results showed that NBI inhibited melanogenesis by decreasing MITF, TRP1, and tyrosinase expression.

### 2.6. Effects of NBI on Signal Transduction Pathways in B16F10 Cells

Activated Akt/GSK-3β and MEK/ERK signals have been reported to degrade MITF [36]. Thus, we investigated whether NBI could activate Akt/GSK-3β and MEK/ERK signaling. We corroborated that ERK and GSK3β phosphorylation were induced following treatment with NBI; however, β-catenin was not degraded (Figure 7).

### 2.7. Effects of NBI on Signal Transduction Pathways in B16F10 Cells

The effect of NBI on melanogenesis through Akt/GSK-3β and MEK/ERK signals in B16F10 cells was investigated. The melanin content confirmed by co-treatment with NBI and Akt/GSK-3β-specific (LY294002) or MEK/ERK-specific (PD98059) inhibitors. As shown in Figure 8, α-MSH-induced melanogenesis was attenuated by NBI treatment, and treatment with each of the specific inhibitors restored the reduction of melanin by NBI.

### 2.8. Effect of NBI on Melanin Production in a Reconstructed Human 3D Skin Model

F-M staining was performed to observe the change in melanogenesis by NBI in a reconstructed human 3D skin model (Figure 9A).

UVB stimulation increased melanin production in 3D human skin and markedly reduced melanin contents by NBI. The UVB-irradiated group presented increased epidermal thickness compared with the normal group, whereas the NBI- and arbutin-treated groups presented a significant decrease in skin thickness (Figure 9B). As a result of quantifying total melanin production, the amounts of melanin were dose-dependently reduced following treatment with NBI (Figure 9C). The rate of inhibition with NBI was 11.8 and 18.8% at 10 and 50 µM, respectively, compared with the UVB-treated group. Inhibition of melanogenesis was equivalent to or greater than that of arbutin 100 µM (16%). These results suggest that NBI has an anti-melanogenic effect on human skin.

## 3. Discussion

Efforts to identify active substances with skin-whitening and anti-aging effects are increasing due to societal demands [37]. Therefore, studies on the various effects of flavonoids and flavonoid glycosides isolated from natural resources, including their anti-oxidative, anti-viral, anti-aging, and whitening effects, are being carried out [38,39,40].

Puerarin, daidzein, and genistein, which are contained in the aerial part of *P. lobata*, have been reported to exert anti-melanogenic effects via MITF [41,42]. In previous studies, the effects of the aerial part of *P. lobata* on the expression of melanin-related genes including MITF, TYR, and TRP-1 have been reported [22]. The active compounds and their inhibitory mechanisms in GALM-DC have not yet been revealed. Therefore, we confirmed the effects and mechanism of action of NBI on melanogenesis.

The B16F10 cell line common widely used to study melanogensis and depigmentation [43]. In cell culture, B16 cells have a reduced ability to synthesize melanin as subculture progresses [44]. There is a need for stimulation of hormones that can induce the synthesis of melanin [45]. α-MSH stimulates tyrosinase activity and induces the synthesis of eumelanin by acting on TRP-1 and TRP-2, enzymes involved in melanin synthesis [46]. The tyrosinase enzyme is a key enzyme in the synthesis of melanin pigment in melanocytes and is absolutely necessary for the synthesis of eumelanin and pheomelanin [47]. Tyrosinase inhibition is important in studies to inhibit melanin synthesis [48]. The melanin content and cellular tyrosinase activity of GALM-DC were confirmed in B16F10 cells that induced melanogenesis by α-MSH stimulation. GALM-DC markedly and dose-dependently inhibited cellular tyrosinase activity and melanin content in B16F10 cells (Figure 2). GALM-DC treatment had no inhibitory effect when tested in the cell-free mushroom tyrosinase system (data not shown). Our results were similar to those of Park, et al. [49]. These results suggest that GALM-DC does not have a direct tyrosinase inhibitory activity and that activity may be degraded through an indirect effect on tyrosinase.

Reactive oxygen scavenging reaction by antioxidants is effective in inhibiting melanin production [50]. DPPH radical scavenging and SOD-like activity were measured to confirm antioxidant activity. DPPH radical scavenging method is a method that is frequently used to measure the antioxidant capacity of flavonoid compounds and aromatic amine compounds [51]. SOD is an enzyme that catalyzes the conversion of oxygen radicals harmful to cells into hydrogen peroxide and protects the body from free radicals [52]. Therefore, the anti-oxidative capacity of GALM-DC, based on the results of DPPH radical scavenging and SOD-like activity, is expected to lead to improved anti-melanogenic effects (Figure 3).

NBI (4′,7-dihydroxy-3′-(3-methyl-2-butenyl)isoflavone) is an isoflavone mainly isolated from Fabaceae family species such as *Psoralea corylifolia* and *Erythrina excelsa*, but we isolated it from GALM [53,54]. NBI is one of increasing interest because it has excellent anti-inflammatory, anti-oxidant and anti- tumor activities [26,27,28].

Melanin synthesis inhibitory activity and molecular mechanism of NBI was investigated in α-MSH-stimulated B16F10 cells. NBI significantly suppressed the melanin content and cellular tyrosinase activity in a dose-dependent manner (Figure 4). These results suggest that NBI reduces intracellular melanin synthesis through down-regulation of tyrosinase activity. To determine the mechanisms of NBI on melanogenesis, we confirmed the expression of NBI melanin-related genes and proteins. NBI treatment reduced the expression of TRP1, tyrosinase and MITF in a dose dependent manner (Figure 6). Previous studies have reported that expression of tyrosinase, TRP-1 and TRP-2 is reduced through MITF expression inhibition [8,41]. These results indicate that NBI contributes to melanin formation inhibition through the down-regulation of MITF in B16F10 cells.

To elucidate how NBI hinders tyrosinase, α-glucosidase activity was investigated. NBI showed better inhibition of α-glucosidase than the positive control acarbose (Figure 5). Tyrosinase is converted to the active tyrosinase when the sugar is modified by α-glucosidase [55]. α-glucosidase inhibitors rapidly fold tyrosinase and transfer it to the melanosome in a copper-free, inactivated form, inhibiting melanin synthesis [56,57,58]. NBI is thought to inhibit the glycosylation reaction of tyrosinase by inhibiting the glucose trimming process required for its post-translation activation.

Melanin contents can be increased by the stimulation of α-MSH. α-MSH combine melanocortin receptor 1 (MC1R) and activates adenylate cyclase, a signaling protein, to increase cyclic AMP (cAMP) [59]. cAMP induces expression of MITF, an important transcription factor for melanogenesis, by down-regulating tyrosinase, TRP-1 and 2 expression [8]. MITF is an intracellular transcription factor important for melanin synthesis and transport [60]. Various studies have been conducted to explain the mechanisms regulating melanin synthesis [42,61,62]. Activation in the MEK/ERK pathway decreases melanin synthesis because it affects MITF activity regulation and stability through phosphorylation [63]. Activated Akt/GSK-3β signaling down-regulates MITF to reduce transcription of tyrosinase and TRP-1 [64]. Previous reports have indicated that betulinic acid and nicotinic acid hydroxamate reduce melanogenesis via regulation of the ERK and AKT pathways [23,36].

To elucidate the mechanisms underlying NBI-induced depigmentation, changes in the activation/phosphorylation of GSK-3β and ERK were examined through western blot analysis. Phosphorylation of GSK-3β and ERK increased significantly following NBI treatment in a dose-dependent manner (Figure 7). NBI was treated in B16F10 with Akt/GSK-3β-specific (LY294002) or MEK/ERK-specific (PD98059) inhibitors to measure melanin content. As shown in Figure 8, melanin production induced by α-MSH was reduced by NBI treatment, and both inhibitors restored NBI-induced melanin reduction. In this study, NBI ameliorated α-MSH-induced melanogenesis through the modulation of Akt/GSK-3β and MEK/ERK signaling pathways in B16F10 cells.

To assess the anti-melanogenesis of NBI was evaluated in a reconstructed human 3D skin model (Neoderm-ME). Evaluation of melanin contents and F-M staining showed that NBI also reduced melanin synthesis in a UVB irradiation-stimulated reconstructed human 3D skin model (Figure 9).

In conclusion, to determine the inhibitory effect of GALM-DC on melanogenesis, melanin contents and cellular tyrosinase activity were investigated. GALM-DC dose-dependently inhibited melanogenesis and cellular tyrosinase activity. Furthermore, we identified NBI as an active compound in GALM-DC and assessed the effects of NBI on melanogenesis as well as its mechanisms of action. NBI is thought to effectively inhibit melanogenesis by inhibiting tyrosinase and TRP-1 through inactivation of the transcription factor MITF. It also suggests that NBI induces the phosphorylation of GSK and ERK, and subsequently decreases melanin synthesis. These results suggest that NBI may be a useful depigmentation compound, and a new alternative in the medical and cosmetics industries.

## 4. Materials and Methods

### 4.1. Materials

Dulbecco’s Modified Eagle Medium (DMEM), antibiotics, phosphate-buffered saline (PBS), fetal bovine serum (FBS), RIPA buffer, and TaqMan RNA-to-CT-1-Step Kit were obtained from Thermo Fisher Scientific (Grand Island, NY, USA). Acarbose, dimethyl sulfoxide (DMSO), L-DOPA, 2,2-diphenyl-1-picrylhydrazyl (DPPH), MTT, α-glucosidase, α-MSH, LY294002, and PD98059 were purchased from Sigma-Aldrich (St. Louis, MO, USA). The SOD Assay Kit-WST was purchased from Dojindo Laboratories (Kumamoto, Japan). The Easy-blue RNA extraction kit was obtained from Intron Biotechnology (Seongnam, Korea).

### 4.2. Extraction and Isolation

The dried and pulverized aerial part of *P. lobata* (10 kg) were extracted under reflux with 30% ethanol at 90 °C for 3 h. The solvent was removed under reduced pressure to give a residue (2.02 kg), which was suspended in distilled water (DW) and partitioned with ethyl acetate (EtOAc) to obtain the EtOAc extract of the aerial part of *P. lobata* (GALM, 145 g). GALM, activated carbon and EtOAc were suspended at a ratio of 1:1.5:7, filtered, and then the solvent was removed under reduced pressure to obtain a decolored extract (GALM-DC, 58 g). Then, GALM-DC was suspended in DW and partitioned with CHCl_3_ to give the CHCl_3_ extract (10.1 g). The CHCl_3_ extract was subjected to column chromatography (CC) on silica gel eluting with a CHCl_3_-MeOH gradient (18:1 to 0:1, *v*/*v*) to give ten fractions (Fr.C1-Fr.C10). Fr.C7 (320 mg) was subjected to a silica gel CC, with a gradient hexane-EtOAc (3:2, *v*/*v*) as the solvent to gain seven subfractions (Fr.C7.1-Fr.C7.7). Repeated silica gel CC (hexane-EtOAc = 3:1, *v*/*v*) of Fr.C7.4 (28 mg) afforded a light yellow powder (21 mg) identified as the prenylated isoflavonoid neobavaisoflavone (NBI, Figure 1) by its ^1^H- and ^13^C-NMR spectral data (Varian Unity Inova AS 400, Varian, Palo Alto, CA, USA). 1H-NMR (MeOD, 400 MHz) δ: 8.03 (1H, s, H-2), 8.01 (1H, d, *J* = 8.8 Hz, H-5), 6.92 (1H, dd, J = 8.8, 2.4 Hz, H-6), 6.81 (1H, d, J = 2.0 Hz, H-8), 6.80 (1H, d, J = 8.4 Hz, H-5′), 7.20 (1H, d, J = 2.0 Hz, H-2′), 7.15 (1H, dd, J = 8.4, 2.0 Hz, H-6′), 5.35 (1H, t, J = 7.6 Hz, H-2″), 3.32 (2H, overlapped, H-1″), 1.71 (3H, s, H-4″), 1.71 (3H, s, H-5″); ^13^C-NMR (MeOD,100 MHz) δ: 153.2 (C-2), 122.9 (C-3), 176.9 (C-4), 127.4 (C-5), 115.1 (C-6), 163.3 (C-7), 101.9 (C-8), 158.4 (C-9), 116.9 (C-10), 124.9 (C-1′), 127.2 (C-2′), 128.0 (C-3′), 155.0 (C-4′), 114.4 (C-5′), 130.1 (C-6′), 28.0 (C-1″), 122.6 (C-2″), 131.7 (C-3″), 24.6 (C-4″), 16.6 (C-5″).

### 4.3. Cell Culture and Cell Viability Assay

B16F10 murine melanoma cells were obtained from the ATCC (Manassas, VA, USA), and cultured in DMEM supplemented with 10% FBS and 1% antibiotics in 5% CO_2_ at 37 °C. B16F10 cells (5 × 10^3^ cells/well) were seeded into a 96-well plate and incubated for 24 h. Cells were treated with GALM-DC or NBI (0–100 µg/mL or µM) for 48 h. The medium was suctioned, then 1 mg/mL MTT solution was added, and the cells were incubated for 3 h in a dark room at 37 °C. After that, formazan was dissolved in DMSO and absorbance was read at 540 nm using a microplate reader (BioTek, Winooski, VT, USA).

### 4.4. Melanin Content Determination

B16F10 cells were plated in 6-well plates at 5×10^4^ cells/well. The following day, cells were incubated with a combination of 100 nM α-MSH and GALM-DC or NBI for 2 days. Cells were washed twice with PBS and lysed in 400 µL of 1N NaOH for 1 h at 95 °C. Absorbance was measured at 405 nm and melanin content was calculated as a percentage from α-MSH-treated controls. The specific inhibitor (PD98059 or LY294002) was pretreated to cells for 1 h, then NBI was added to cells.

### 4.5. Measurement of Cellular Tyrosinase Activity

B16F10 cells were incubated with a combination of 100 nM α-MSH and GALM-DC or NBI for 2 days. The pellet was obtained by washing with PBS. The cells were lysed with 0.1 M sodium phosphate buffer (pH 6.8) containing 5 mM EDTA, 1% Triton X-100, and 0.1% phenylmethylsulfonyl fluoride (PMSF) in ice for 30 min. Lysates were centrifuged at 14,000 rpm for 25 min to obtain supernatants, followed by protein quantification by the Bradford method. The reaction mixture of 50 µg of protein, 0.1% L-DOPA and 100 mM SPB was incubated at 37 °C for 1 h, and the tyrosinase activity was estimated by measuring the absorbance at 475 nm.

### 4.6. Measurement of MITF, TRP-1, and Tyrosinase mRNA Expression

The mRNA was quantified by ND-1000 spectrophotometer (NanoDrop Technology, Wilmington, DE, USA). An mRNA sample (300–500 ng, primer and TaqMan RNA-to-Ct 1-Step Kit were applied according to the manufacturer’s instructions. The assay ID of the primers are the following: Mitf-Mm00434954_m1, TRP-1-Mm00453201_m1, Tyrosinase-Mm00495818_m1 and GAPDH-Mm99999915_g1. Real-time PCR was performed at 48 °C for 15 min, at 95 °C for 10 min followed by 40 cycles at 95 °C for 15 s and 60 °C for 1 min on an ABI Step One Plus system (Applied Biosystems, Foster City, CA, USA). Relative quantification of mRNAs was calculated by the ΔΔCT method.

### 4.7. Western Blot Analysis

Proteins were obtained using RIPA buffer containing protease inhibitors and then quantified. The proteins (40 µg) were electrophoresed by 8% SDS-PAGE gels and transferred to nitrocellulose. The membranes were blocked with 5% skim milk in PBST (PBS containing 0.1% Tween 20) at RT for 1 h, washed with PBST, incubated with primary antibodies (1:1000) for 16 h at 4 °C, washed with PBST, incubated with HRP-conjugated secondary antibodies for 1.5 h at RT. Western blotting was performed using the following antibodies; MITF antibody(sc-56725, Santa Cruz Biotechnology, Santa Cruz, CA, USA), TRP-1(sc-166857, Santa Cruz Biotechnology), tyrosinase (sc-20035, Santa Cruz Biotechnology), p-ERK (4370, Cell Signaling Technology, Beverly, MA, USA), p-GSK3β (5558, Cell Signaling Technology), β-Catenin (9562, Cell Signaling Technology), and GAPDH (5174, Cell Signaling Technology). Anti-mouse and anti-rabbit IgG antibodies were from Santa Cruz Biotechnology. The protein-antibody complex was visualized by an ECL system. Bands were quantified using the FluorChem E system image analyzer (Cell Biosciences, Santa Clara, CA, USA). GAPDH was used as an internal control. For quantify, images of protein bands were measured using Image J software (NIH, Bethesda, MD, USA).

### 4.8. α-Glucosidase Inhibition Assays

The α-glucosidase enzyme inhibition assay was performed according to the method described by Si et al. [65]. Briefly, 10 µL of NBI or acrbose at various concentrations (31.3, 62.5, 125, 250, 500, and 1000 µM or µg/mL), 20 µL of α-glucosidase (0.5 unit/mL), and 120 µL of 0.1 M phosphate buffer (pH 6.9) were mixed. After incubating at 37 °C for 15 min, 20 µL of 5mM 4-nitrophenyl-α-d-glucopyranoside was put into substrates, which were incubated for an additional 15 min. The reaction was terminated by the addition of 80 µL of 200 mM sodium carbonate (Na_2_CO_3_). The absorbance was measured at 405 nm. Acarbose (an α-glucosidase inhibitor) was used as a positive control. The enzyme inhibitory rate was calculated as follows.

### 4.9. DPPH Free Radical Scavenging Activity

One hundred µL of the diluted GALM-DC and 100 µM DPPH solution in methanol were added respectively in a 96-well plate. The mixture was incubated for 30 min at RT in dark conditions then the absorbance was measured at 520 nm. Vit C was used as a positive control.

### 4.10. Superoxide Radical Scavenging Activity

The SOD activity was determined by a SOD Assay Kit-WST (Dojindo, Tokyo, Japan) according to the manufacturer’s instructions. Briefly, 20 µL of GALM-DC was added with 200 µL WST solution and 20 µL enzyme solution in a 96-well plate. After incubation at 37 °C for 20 min, absorbance was measured at 450 nm.

### 4.11. Culture of Reconstructed 3D Skin Model

A reconstructed human 3D skin model (Neoderm-ME) was obtained from Tegoscience Co. (Seoul, Korea). In brief, Neoderm-ME was transferred to a 12 well plate and stabilized at 37 °C in 5% CO_2_ for one day. Treatment with vehicle or NBI or the positive control arbutin for 1 h before UVB (50 mJ/cm) irradiation for 8 days. The medium was changed every day and the plate was incubated at 37 °C and 5% CO_2_. Then, 3D human skin tissue was dissolved in 300 µL of 1N NaOH for 1 h at 95 °C to measure melanin content and subjected to Fontana-Masson (F-M) staining.

### 4.12. Fontana-Masson Staining

To observe the degree of skin pigmentation, F-M staining was performed. 3D human skin tissue blocks were fixed with 4% formalin for 18 h and embedded in paraffin. Sections cut to 4 µm thickness were stained using the F-M staining kit from IHC World (Woodstock, MD, USA). Briefly, each slide is deparaffinized and rinsed with distilled water. Fontana silver nitrate working solution at 60 °C for 1 h in a dark chamber. After rinsing in distilled water, the slides were placed in gold chloride working solution for 1 min, rinsed in distilled water, and then in 5% sodium thiosulfate solution for 1min. Rinse again with distilled water, the slides were counterstain with nuclear fast red solution for 3min. Rinse thoroughly in distilled water twice. After dehydration and mounted on a slide, the sections were observed with a Leica phase-contrast microscope (Leica Microsystems, Wetzlar, Germany). The stained slide was observed under a microscope to measure epidermal thickness. The thickness of the epidermis was evaluated using the average of three measurements from each sample.

### 4.13. Statistical Analysis

Data were expressed as mean ± SD. Significant differences were compared using repeated measures ANOVA followed by the Tukey’s multiple range test. Statistical significance was defined as *p* < 0.05. All statistical analyses were performed using GraphPad Prism version 5.0 software (GraphPad Software Inc., La Jolla, CA, USA).

## Figures and Tables

**Figure 1 molecules-25-02683-f001:**
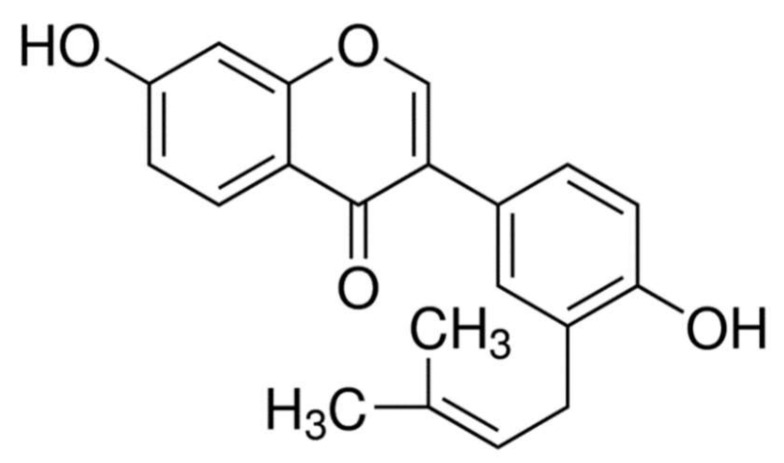
Chemical structure of neobavaisoflavone (NBI) isolated from extracts (GALM-DC) from the decolorized aerial part of *Pueraria lobata*.

**Figure 2 molecules-25-02683-f002:**
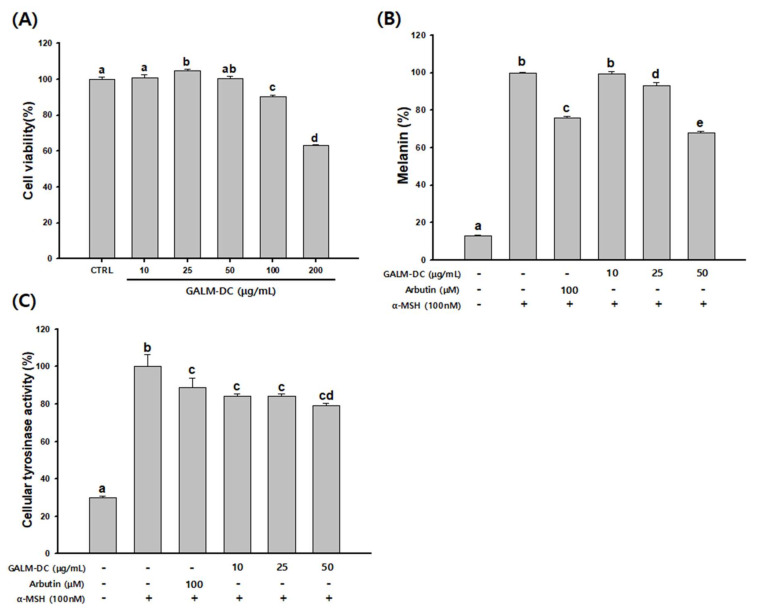
Anti-melanogenesis effects of GALM-DC on B16F10 cells. Cells were cultured with GALM-DC (10–50 µg/mL) for 48 h. (**A**) Cytotoxicity, (**B**) melanin contents, and (**C**) cellular tyrosinase activity were measured. Arbutin was used as a positive control. Data are presented at the mean ± SD. Values with different letters (a, b, c, d, e) are significantly different one from another (one-way ANOVA followed by Tukey multiple range test, *p* < 0.05).

**Figure 3 molecules-25-02683-f003:**
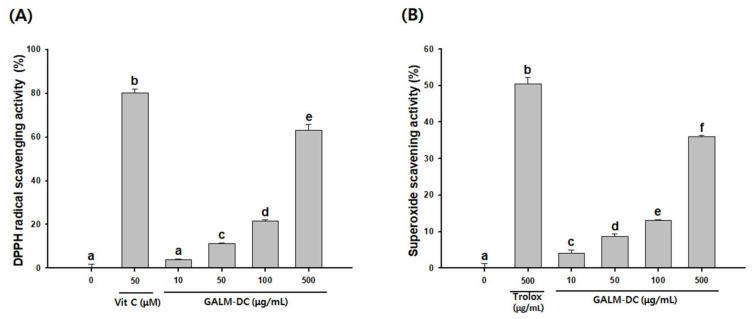
Anti-oxidant activities of GALM-DC on various radicals determined by (**A**) DPPH radical scavenging activity and (**B**) superoxide scavenging activity. Vitamin C (**A**) and trolox (**B**) were used as positive controls. Data are presented as mean ± SD. Values with different letters (a–f) are significantly different one from another (one-way ANOVA followed by Tukey’s multiple range test, *p* < 0.05) sarcopenia.

**Figure 4 molecules-25-02683-f004:**
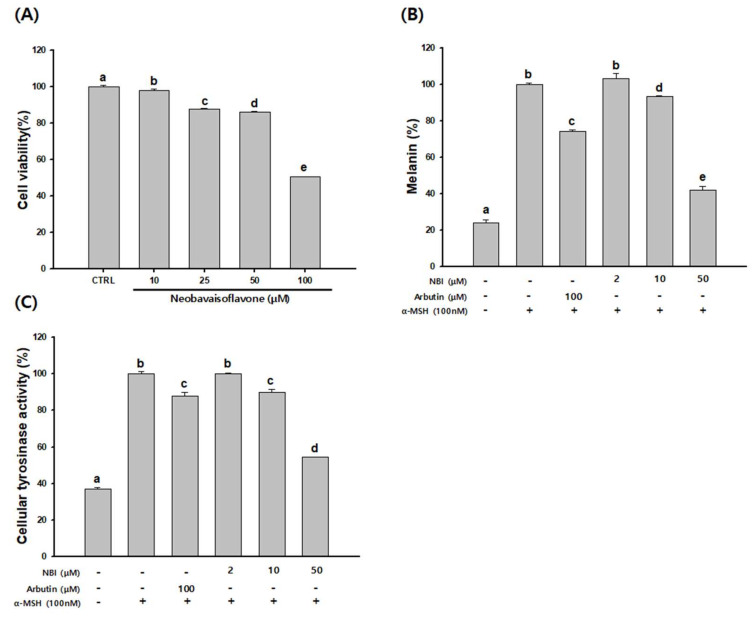
Effects of NBI on anti-melanogenesis in B16F10 cells. Cells were cultured with NBI (2–50 µM) for 48 h. (**A**) Cytotoxicity, (**B**) melanin contents, and (**C**) cellular tyrosinase activity were measured. Data are presented as mean ± SD. Values with different letters (a, b, c, d, e) are significantly different one from another (one-way ANOVA followed by Tukey’s multiple range test, *p* < 0.05).

**Figure 5 molecules-25-02683-f005:**
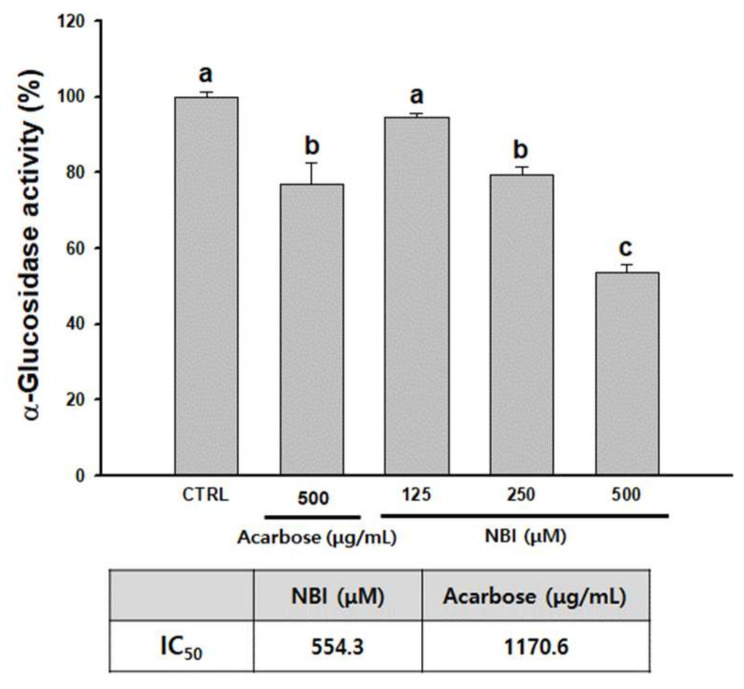
Inhibitory effects on α-glucosidase activity. Each percentage value representing α-glucosidase activity is reported relative to that of the control. Data are presented as mean ± SD. Values with different letters (a, b, c) are significantly different one from another (one-way ANOVA followed by Tukey’s multiple range test, *p* < 0.05).

**Figure 6 molecules-25-02683-f006:**
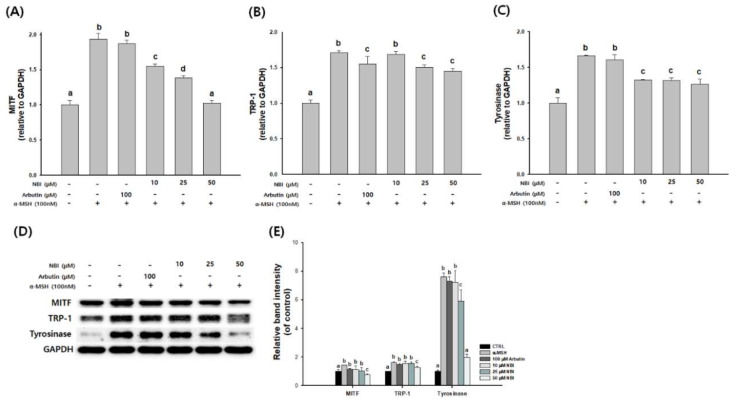
Effects of NBI on the expression of melanogenesis-related mRNA and proteins. B16F10 cells were treated with α-MSH and NBI at the indicated concentration. (**A**) MITF mRNA levels, (**B**) TRP1 mRNA levels, (**C**) tyrosinase mRNA levels, (**D**) MITF, TRP1, and tyrosinase protein expression levels, and (**E**) The graph indicates the expression level against GAPDH expression level. Data are presented as mean ± SD. Values with different letters (a, b, c, d) are significantly different one from another (one-way ANOVA followed by Tukey’s multiple range test, *p* < 0.05).

**Figure 7 molecules-25-02683-f007:**
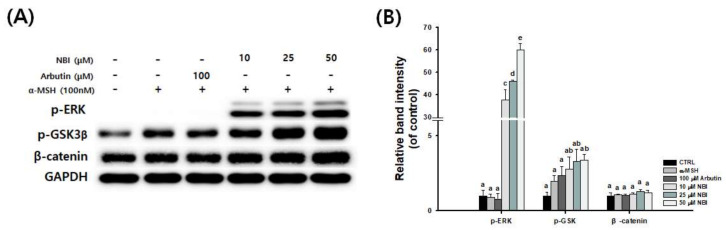
Effect of NBI on GSK3β- and ERK-dependent signaling in B16F10 cells. B16F10 cells were treated with α-MSH and NBI at the indicated concentration. (**A**) p-ERK, p-GSK, and β-catenin protein expression levels, and (**B**) The graph indicates the expression level against GAPDH expression level. Data are presented as mean ± SD. Values with different letters (a, b, c, d, e) are significantly different one from another (one-way ANOVA followed by Tukey’s multiple range test, *p* < 0.05).

**Figure 8 molecules-25-02683-f008:**
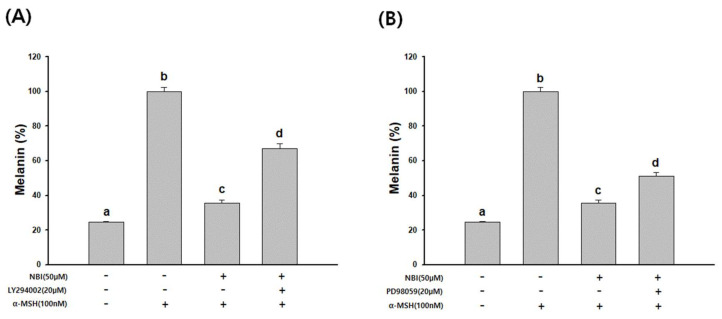
Effects of NBI on the regulation of Akt/GSK-3β and MEK/ERK signaling pathways. Melanin contents were evaluated. B16F10 cells were co-treated with NBI and specific inhibitors of (**A**) MEK/ERK (PD98059) and (**B**) Akt/GSK-3β (LY294002) signaling. Cells were stimulated with α-MSH (100 nM) and pretreated for 1 h in the absence (−) or presence (+) of Akt/GSK-3β and MEK/ERK pathway-specific inhibitor, and then cultured without (−) or with (+) 50 µM NBI for 48h. Data are presented as mean ± SD. Values with different letters (a, b, c, d) are significantly different one from another (one-way ANOVA followed by Tukey’s multiple range test, *p* < 0.05).

**Figure 9 molecules-25-02683-f009:**
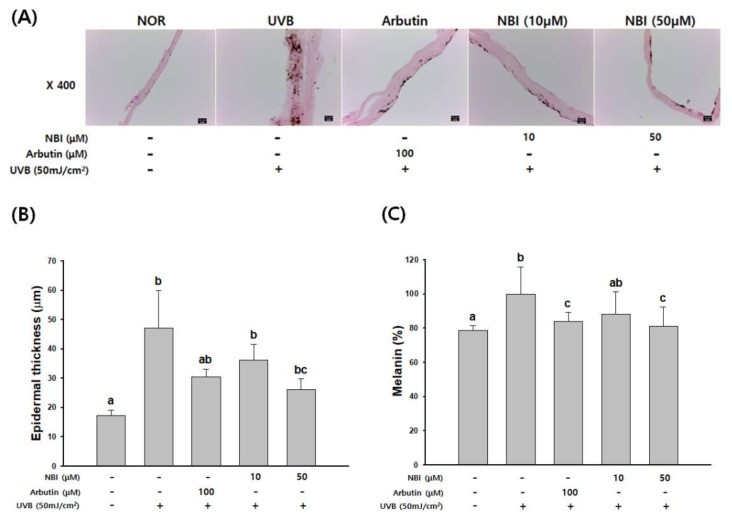
The inhibitory effect of NBI on melanin production in a reconstructed human 3D skin model skin model. (**A**) Fontana-Mason (F-M) staining of tissue sections. (**B**) Measurement of epidermal thickness. (**C**) Melanin content in a reconstructed human 3D skin model. Data are presented as mean ± SD. Values with different letters (a, b, c) are significantly different one from another (one-way ANOVA followed by Tukey’s multiple range test, *p* < 0.05).

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
