# Peer review of "Neobavaisoflavone Inhibits Melanogenesis through the Regulation of Akt/GSK-3β and MEK/ERK Pathways in B16F10 Cells and a Reconstructed Human 3D Skin Model"

_molecules, 2020, doi:10.3390/molecules25112683_

Round 1

Reviewer 1 Report

In this work, the neobavaisoflavone (NBI) was identified as an active compound from the decolorized aerial part of P. lobata. The authors performed a series of experiments to demonstrate that the NBI effects on melanogenesis and mechanisms of action in melanocytes.  From these experiments, the authors suggested that  NBI has potential as a skin-whitening agent for hyperpigmentation. Although the novelty of the discovery is evident and experiments performed in this work are standard, the scientific quality of data presentation has to be improved. 1) The logic of standard biochemistry, which was applied in the Results section has to be explained in the Introduction. Currently, this link is missed in the manuscript.
2) All abbreviations in the text have to be explained in a separate abbreviations paragraph.
3) The effect of acarbose (line 112) has to be explained and properly-referenced.
4) The reference has to be provided to the sentence 'Activated Akt/GSK-3β and MEK/ERK signals have been reported to degrade MITF' (Line 136).
5) In Lines 243-244 exact names of purchased antibodies should be shown.

Author Response

Dear Reviewer

It is with excitement that we were granted the opportunity to resubmit to you a revised version of manuscript Title: ‘Neobavaisoflavone Inhibits Melanogenesis Through the Regulation of Akt/GSK-3β and MEK/ERK Pathways in B16F10 Cells and a Reconstructed Human 3D Skin Model’. We have highlighted the changes in the text for the ease of your review.

Best regards,

Reviewer 2 Report

This manuscript describes interesting issues regarding the effect of neobavaisoflavones on melanogenesis.  However, this manuscript needs major corrections.

Detailed comments are provided below

Introduction

  1. The Introduction lacks information about the MITF factor and signaling pathways. These proteins were examined in the present work and their role in the melanogenesis process should be indicated (this information appears later in the discussion). In my opinion, this information should be transferred to introduction, it will help the reader to understand the content.
  2. Line 34, color of the skin and hair d functions?????
  3. Line 63, this should be – Extracts (GALM-DC) from the decolorized………
  4. Line 64, not Neobaibaisoflavones but neobavaisoflavones
  5. There is no clear hypothesis and research goal, please, provide these
  6. Line 69, this last sentence is rather a conclusion not the aim of the study and should be provided in Conclusion paragraph (section).

Results

  1. Line 72, After treatment with GALM-DC at 0 , 10… (This is rather not precise information, concentration 0 means that this factor does not occur and it is a control group).
  2. Line 74, the same suggestion (In B16F10 cells, it proceeded at a non-toxic concentration of GALM-DC (0-50); and what about 100 µg/ml it was also not cytotoxic concentration.
  3. Captions to Figure 1. The authors should mention what Arbutin is?, and underline that this is a positive control.
  4. 2B- please, correct description of y axis – it should be superoxide scavenging
  5. Line 86, The SOD activites of GALM-DC at 100 and 500 µg/ml were 13.1, 2 and 36% respectively, ( 34.2 or 36%??????), while that of 500 µg/ml ( rather µM, such a unit is given on the graph, Fig. 2)
  6. The paragraph -Effects of NBI on anti-melanogenesis in B16F10 cells should be rewritten.

Lines 96-99, As shown in Figure. 3C, the vehicle and α-MSH-treated groups produced 9.2 and 38 µg/1.5 × 106 cells of melanin, respectively, and the amount of melanin induced by α-MSH increased. Furthermore, α-MSH-induced melanin production was inhibited in a dose-dependent manner; 35.6 and 16 µg/1.5 × 106 cells of melanin were detected following treatment with 25 and 50 µM NBI, respectively. This description is incomprehensible and not consistent with the data presented in diagram C (Figure 3). In addition, concentration 25 µM is not marked on graphs C and D (Figure 3)

  1. Line 111, Authors should indicate the correlation between α-glucosidase activity and tyrosinase maturation
  2. Why did the authors use different NBI concentrations to determine α-glucosidase activity than those used in the whole experiment?
  3. Lines 120-121; cells were treated with NBI stimulation with α-MSH (???? This is not clear, mayby it should be after stimulation with α-MSH)
  4. Line 123; The inhibition rates of MITF expression of MITF (delete repetition) were….
  5. 5D, On the presented blots, it is difficult to notice differences in TRP-1 protein expression between the α -MSH treated group and the NBI treated group (at 10 and 25µM concentration). The Authors should make a quantitative analysis. In the chapter materials and methods, the Authors inform that they have done such an analysis, however, there is no such data in the description of the results. The presented graphs show the quantitative analysis of mRNA expression.
  6. The same applies to figure 6. The authors should quantify the expression of the proteins tested. Please change the description of these results and of the figure 6 as well.
  7. As the authors explain the lack of p-ERK expression in control and after MSH. After all, other processes take place in the cells in which the studied signal pathways can be involved?
  8. Line 156-157; ………UVB-induced melanin production was monitored by using F-M staining

Discussion

  1. Laconic discussion, requires extension. A lot of results, which only briefly were discussed with several literature items.

Materials and methods

  1. This chapter was written very briefly, as if the authors were trying to shorten the previously written work. They use many abbreviations that are not understandable to the reader. There is a great deal of lack of data on the reagents used and the analyzes carried out. This chapter requires major correction.

Extraction and isolation

Line 246-247; The dried and pulverized aerial part of P. lobata (10 kg) was extracted three times with ethanol  (in what concentrations? how long? ) under reflux.

EtOAc, GALM-DC- explain the abbreviations

Line 247-250; vague (misleading) description, From this description it follows that GALM is an extract suspended in EtOAc, and after adding activated carbon it is again suspended in EtOAc (?????)

Cell culture and cell viability assay

Use superscript and subscript (CO2; 5x106)

Line 269-270, The medium was suctioned and 1 mg/mL MTT solution was added  and incubated  (in dark) for 3 h ( in what temperature?)

Melanin content determination

Line 273-274; The following day, cells were  incubated with 100 nM α-MSH and GALM-DC or NBI for 2 days. This should be better: The following day, cells were  incubated with a combination of 100 nM α-MSH and GALM-DC or NBI for 2 days.

Line 273, Use superscript ( 5x104)

Measurement of cellular tyrosinase activity

Lines 280-281, These ( use Lysates) were centrifuged at 14,000 rpm for 25 min (at 4ᵒC) to obtain supernatants, followed by protein quantification (How?, Bradford method?)

Lines 282-283, The reaction mixture of 50 µg of protein, 0.1% L-DOPA and 100 mM SPB was incubated at 37℃  for 1 h, and the tyrosinase activity was estimated by measuring the absorbance at 475 nm.

 Measurement of MITF, TRP-1, and tyrosinase mRNA expression

  • The Authors should provide sequences of used primers
  • How was the level of mRNA calculated? (method?)

Western blot analysis

  • The Authors should give the name of used antibodies, provide the basic data concerning the primary antibodies (mono or polyclonal, mouse, rabbit or other)

             α-Glucosidase inhibition assays

  • Briefly, 10 µL of test samples ( it is better to write NBI) at various concentrations…..
  • pNPG- explain the abbreviation
  • line 304, Na2CO3
  • line 304, Acarbose (α-glucosidase inhibitor) was used as a positive control

       DPPH and SOD free radical scavenging activities

  • Why were DPPH and SOD free radical scavenging activities estimated only in the GALM-CD extract but not in NBI?
  • Line 308, 100 µL of the diluted sample (GALM-CD?) and 100 µM DPPH solution in methanol were added (where? In a 96-well plate???)

  Superoxide radical scavenging activity

  • line 314, Briefly, 20 µL of the sample ( GALM-CD) was added with 200 µL WST solution 315 and 20 µL enzyme solution (where?? in a 96-well plate???)

Culture of reconstructed 3D skin model

  • CO2
  • What is arbutin?, explain
  • How was the malanin content estmated on 3D skin model?

Fontana-Masson staining

  • Line 326, 3D human skin tissue blocks were fixed with formalin (in what concentration?, how long?) and what about rinse?? and embedded in paraffin. Sections cut to 4 µm thickness were  stained using the F-M staining kit from IHCWORLD (GA, USA) – please, describe this F-M staining in details .
  • How was the epidermal thickness estimated??

Author Response

(The authors gave the same response as above.)

Reviewer 3 Report

Title: Neobavaisoflavone Inhibits Melanogenesis Through the Regulation of Akt/GSK-3β and MEK/ERK Pathways in B16F10 Cells and a Reconstructed Human 3D Skin Model by Da Eun Kim et al.

Research Article Id:  molecules-811800

The authors have identified neobavaisoflavone (NBI) as an active compound from the decolorized aerial part of P. lobata (GALM-DC), investigated its effects on melanogenesis, and elucidated its mechanisms of action in melanocytes. However, to consider this manuscript needed some corrections to strengthen the results.

I found several short falls in this manuscript as follow:

  • English is not clear enough that, intended meaning can be gleaned. I would recommend, authors ask a colleague who is a native English speaker to help polish the article first as well as several typographical errors should be corrected.
  • The abstract and Introduction section is very vaguely written. It needs re-writing and improvement. Several recent potentials published articles on the subject are not included while presenting the state of art in this section. The authors need to emphasize importance of this work over the earlier reports.
  • In introduction section, authors state that, due to the inherent color of the aerial part of P. lobata, there is a limit to its utilization in the cosmetic industry. Therefore, in the present study, they decolored the aerial part of P. lobata using activated carbon. Extracts from the decolorized aerial part of P. lobata (GALM-DC) were found to contain Neobaibaisoflavones (NBI) and seven other compounds. NBI are isolated from the Fabaceae families such as Psoralea corylifolia, Erythrina excelsa, and Erythrina senegalensis, and are one of the isoflavones [21,22]. To make it more clear, authors should mention the name of few of compounds here.
  • To make this work more interesting to reader, authors could explain the reason for choosing BI6F10 cell line? and also name of primers and antibodies (primary and secondary) used in measurement for mRNA expression and western bloating should be indicated in methodology section.
  • Independent t-test analysis is not enough to determine the significance in gene expression analysis so authors should use ANOVA to analyze the significant difference among more than 2 groups rather than t-test.
  • In result section 2.3. effects of NBI on anti-melanogenesis in B16F10 cells, authors showed 100 µM NBI have a significant cell cytotoxicity about 50% of  B16F10 cell death in Figure 3B. However, in next results, 2.4. inhibitory effect of NBI on glucosidase activity section, authors further claims NBI reduced α-glucosidase activity with an IC50 of 554.3 µM in figure 4. Authors need to explain, how this IC50 calculated in table below figure 4?
  • How long the cells are simulated (pre-incubate or post-incubate) with inhibitors prior to NBI treatment?
  • As shown in Figure 1B and C, melanin contents and cellular tyrosinase activity were measured c.a. 70% and c.a. 80% after exposure to GALM-DC. However, in Figure 3c, melanin contents and cellular tyrosinase activity were measured c.a. 40% and c.a. 60% after exposure to NBI which are decreased in value compare with GALM-DC. What’s the reason for that?
  • In the result section 2.3, effects of NBI on anti-melanogenesis in B16F10 cells, paragraphs are not clear to understand, these should rephrase again.
  • In discussion sections, from line numbers 181 to 190, authors speculated that, reactive oxygen scavenging reaction by antioxidants is effective in inhibiting melanin production [35]. Therefore, the anti-oxidative capacity of GALM-DC, based on the results of DPPH and SOD radical scavenging assays, is expected to lead to improved anti-melanogenic effects (Figure 2). Tyrosinase is an important early rate determining enzyme in the melanin synthesis pathway [36]. GALM-DC treatment had no inhibitory effect when tested in the cell-free mushroom tyrosinase system (data not shown). GALM-DC markedly and dose-dependently inhibited cellular tyrosinase activity and melanin content in B16F10 cells (Figure 1B and 1C). These results suggest that GALM-DC does not have a direct tyrosinase inhibitory activity and that activity may be degraded through an indirect effect on tyrosinase. These are disjointed paragraphs.  The paragraphs need to be linked better, so that the manuscript is coherent.

Author Response

(The authors gave the same response as above.)

Round 2

Reviewer 2 Report

The manuscript has been corrected as suggested by the reviewer.

Reviewer 3 Report

Authors' responses are satisfactory and they presented point to pint of all my quarry therefore I recommended this manuscript for publication.